# Clinical Testing of Walking Bleach, In-Office, and Combined Bleaching of Endodontically Treated Teeth

**DOI:** 10.3390/medicina59010018

**Published:** 2022-12-21

**Authors:** Natasa Knezevic, Marija Obradovic, Olivera Dolic, Valentina Veselinovic, Zeljka Kojic, Renata Josipovic, Marijana Arapovic-Savic

**Affiliations:** Department of Dentistry, Faculty of Medicine, University of Banja Luka, 78000 Banja Luka, Bosnia and Herzegovina

**Keywords:** walking bleach, in-office, combined technique, carbamide peroxide, hydrogen peroxide

## Abstract

**Objective.** The causes of internal posteruptive discoloration of teeth are bleeding, necroses, infections, and endodontic filling materials. The aim of this study was to establish the results of bleaching endodontically treated teeth using walking bleach, in-office, and combined techniques, using 30% carbamide peroxide and 35% hydrogen peroxide, as well as the effect of etiological factors, and the time elapsed after endodontic treatment on the success of bleaching. Materials and Methods. The research involved 30 endodontically treated teeth in healthy patients. Retroalveolar X-rays were taken to check the quality of root canal obturation. Endodontic treatment and obturation were carried out on the discolored non-vital teeth without any previous endodontic treatment. Before bleaching, two millimeters of the filling were removed from the root canal and the very entry into the canal was protected with glass ionomer cement. The teeth were divided into three groups, depending on the bleaching technique: walking bleach technique (10 patients), in-office technique (10 patients), and combined technique (10 patients). The teeth were bleached with 30% carbamide peroxide and 35% hydrogen peroxide. The bleaching procedure was repeated in all the patients three times. The color of all teeth was determined based on the Vita Classic guide before and after bleaching. The Χ^2^ square and Kruskal–Wallis tests were used to identify differences in teeth bleaching results. Results. A statistically significant difference (*p* < 0.05) was established between bleaching success and the time elapsed after endodontic treatment. There were no statistically significant differences observed between the bleaching success and etiological factors, bleaching techniques, or bleaching agents. Conclusions. The effectiveness of non-vital tooth bleaching is affected by the time elapsed after endodontic treatment.

## 1. Introduction

Dental dyschromia varies in etiology, location, and severity, and its origin may be related to extrinsic causes, intrinsic causes, or a combination of both. A detailed study of the case must be carried out to make an accurate diagnosis, since the success of the treatment and the accuracy of the results will depend on it [1].

The most common causes of internal posteruptive discoloration of teeth are bleeding, necroses, infections, and iatrogenic factors. An internal change in color may be caused by the application of various irrigants used for the chemical processing of canals, medications, and endo sealers. In the past, this type of discoloration was mainly treated by invasive prosthetic restoration [2,3,4,5].

Internal tooth bleaching is a minimally invasive, conservative, relatively simple, effective, and low-cost method in the treatment of discolored endodontically treated teeth [6,7,8].

Modern dentistry is based on the concept of minimum intervention, that is, on saving dental tissue. This concept specifically refers to tooth discoloration treatment. The bleaching of teeth and restoration of their natural color helps save dental tissue and also provides an alternative solution to crowns [2,3,4].

The intracoronal bleaching procedure uses oxidizing agents within the coronal portion of an endodontically treated tooth to remove tooth discoloration (American Association of Endodontists, 2003) [9]. The bleaching of endodontically treated teeth is based on dentin permeability, allowing the oxidizing agent to penetrate directly into the pigment in the dentin, and eliminate or alleviate the problem of discoloration [9,10]. The oxidizing chemical agent removes intrinsic stains via chromogenic degradation, and by breaking down the larger pigments into smaller ones, the color of the teeth is lightened [1].

Today, the most common dental bleaching agents are hydrogen peroxide and carbamide peroxide, which are used in high concentrations for non-vital teeth, but the mechanism of action remains the same in all cases. That is, oxidation occurs with organic pigments, and the products of decomposition of the chemical agent are used [7,11,12].

The bleaching prognosis is uncertain and depends on several factors, such as the cause of discoloration, bleaching techniques, and agents. The change in color caused by necrosis or trauma is usually good. This is because the discolorations occurring due to the degradation of pulp and blood products usually react positively to the bleaching process. The prognosis of bleaching also depends on the kind of endodontic filling materials, the time when the discoloration occurred, and the time since endodontic treatment [10,13].

The aim of this clinical study was to establish the results of bleaching endodontically treated teeth using the methods of walking bleach, in-office, and combined bleaching techniques, using 30% carbamide peroxide and 35% hydrogen peroxide, as well as the effect of the etiological factors, and the time elapsed after endodontic treatment on the success of bleaching.

The null hypotheses were as follows:There is no difference in clinical efficiency of intracoronal bleaching between the time elapsed after endodontic treatment.There is no difference in clinical efficiency of intracoronal bleaching between different discoloration causes.There is no difference in clinical efficiency of intracoronal bleaching between different bleaching techniques.There is no difference in clinical efficiency of intracoronal bleaching between different bleaching agents.

## 2. Materials and Methods

The study was carried out at the Faculty of Medicine in Banja Luka, Study Program of Dentistry, as part of a research project for a Ph.D. thesis. The research was conducted in full accordance with the 1964 Declaration of Helsinki and its subsequent revisions or comparable ethical standards, whereas the ethical approval for the study was obtained from The Research Committee of the Faculty of Medicine, University of Banja Luka (No: 18-3-692/2012; 11/24/2012). The research was carried out between 2013 and 2014.

The patients signed a consent form for taking part in the study, and were informed of the procedure, multiple-session treatment, and possible side effects or failure of the bleaching procedure.

### 2.1. Study Design

This blind randomized clinical study compared three different methods of intracoronal bleaching. Patients were invited to participate in thorough examinations at the exercises of dental students. The treated teeth were randomly selected. The distribution of bleaching techniques by groups of teeth was carried out by random drawing using Microsoft Excel 2010 (Microsoft, Redmond, WA, USA) from the coding assigned to each participant. There were three groups: the first group was bleached with the walking bleach technique (10 teeth), the second group was bleached with the in-office technique (10 teeth), and the third group was bleached with the combined technique (10 teeth). Two trained operators performed the bleaching treatments. The third operator, who had no contact with the patients, was responsible for conducting the randomization. To ensure double blinding, marks and packaging on bleaching agents were removed, the procedures and the instruments were standardized, and randomization was alphanumerically coded. The statistician received data tabulated in code that did not allow for the identification of the treatment applied to each group.

The research involved 30 endodontically treated teeth in 26 healthy patients. Three teeth were bleached in one patient, two endodontically treated teeth in two patients each, and one tooth was bleached in 23 patients. The patients were selected based on their dental status, namely, the presence of discolorations in non-vital and endodontically treated teeth. 

The inclusion criteria were as follows: male and female patients between 18 and 50 years of age who signed up to participate in this research, good systemic and oral health, and visibly discolored non-vital or endodontically treated incisors, canines, and premolars, with colors darker than A2 on the Vita Classical shade guide. 

The exclusion criteria were as follows: advanced periodontal diseases, teeth with large fillings or large carious lesions that could not be reconstructed using conservative procedures, tooth pain, periapical lesions, bone resorption and systemic diseases, pregnant or lactating, moderate or severe fluorosis, tetracycline stains, orthodontic treatment, orofacial tumors, trauma, or tooth malformation (Figure 1) [14].

### 2.2. Sample Size Calculation

A power analysis was completed for repeated measures of the Wilcoxon test. The output parameters were the noncentrality parameter δ = 2.57, critical t = 1.71, df = 25.32, and total sample size *n* = 24; actual power (Ap) was = 0.8. This study included eight cases per group for a total of twenty-four cases. To compensate for a dropout rate of approximately 15–20%, the required total sample size in this study was 30 cases. A sample size calculation was performed using G*Power^®^ Version 3.1.9.7 (Düsseldorf, Germany).

### 2.3. Independent Variables Tested

The bleaching result analysis was performed based on the time elapsed after endodontic treatment, etiological factor, and results analysis for the bleaching technique and the agent used. 

Based on the time the discoloration occurred, or the time when endodontic treatment was completed, we divided the teeth into those endodontically treated within the past 5 years, and teeth endodontically treated more than 5 years ago.

Depending on the etiological effect of discoloration of endodontically treated teeth, discolorations were divided into endo sealer discoloration, discoloration caused by trauma, discoloration caused by necrosis/infection, and discoloration of unknown etiology. The sealer-induced tooth discoloration included the teeth in which the endodontic filling materials (endo sealers and gutta percha in the cavity) were observed on the cavity walls. The teeth discolored by trauma included the teeth for which a patient cited the history of a fall or blow to the tooth. In the discolorations caused by necrosis/pulp infection, necrotic or infected content was found in the tooth cavity, without any anamnestic data of trauma. The discolored teeth whose cause of discoloration we were unable to establish were classified as discolorations of unknown etiology. Apart from the history of trauma, excess gutta percha and endo sealer were found in the cavities of such teeth. The likely parallel causes of discoloration of this group of teeth could have been trauma, endo sealer and/or necrosis/infection.

Depending on the bleaching technique, the teeth were grouped into those bleached using the in-office technique, walking bleach technique, and combined technique, whereas, depending on the bleaching agent, the teeth were grouped into those bleached with 30% carbamide peroxide, and those bleached with 35% hydrogen peroxide.

The results of bleaching concerning the previous endodontic treatment, the time elapsed after endodontic treatment, and etiological factors are presented in relation to the entire sample.

### 2.4. Preparation of Teeth Prior to Bleaching 

All the patients had a clinical examination. Retroalveolar X-rays were taken to check the quality of root canal obturation. Inadequate obturation was revised. Endodontic treatment and obturation were also carried out on the discolored non-vital teeth without any previous endodontic treatment. Endodontic treatment included the formation of an access cavity, trepanation, odontometry, mechanical instrumentation in the cleaning of root canals, including reamers and files (Kerr, Orange, CA, USA), and a simultaneous root canal irrigation with sodium hypochlorite (Patenting, Belgrade, Serbia), EDTA (Ultradent, South Jordan, UT, USA), and saline solution (Hemofarm, Vršac, Serbia). The teeth were definitively obturated with the AH plus sealer (Dentsply Sirona, Charlotte, North Carolina, USA) and gutta percha points (Spident, Incheon, Korea), using the cold lateral compaction technique. Prior to bleaching, two millimeters of the filling were removed from the root canal, and the very entry into the canal was protected with a liner based on the glass ionomer cement Alfagal Bejz (Galenika, Belgrade, Serbia), to prevent the penetration of the bleaching agents into endodontium or any damage to the periapex, as well as the penetration of endo sealer into the coronal portion of the tooth [15].

### 2.5. Testing/Experimental Procedure

Prior to the treatment, the current color of all teeth was determined based on the Vita Classic guide (Vita Classic, Lumin Vacuum Shade Guide, Vita Zahnfabrik, Bad Säckingen, Germany). The procedure was performed in the daytime under natural light. We checked the color in the middle third area of the labial surface of the anterior central incisors according to the American Dental Association guidelines [16]. Three operators were responsible for evaluating the color of the teeth. The operators were professors of dentistry. Before this research, they attended training on the basic concepts of color and the factors that may influence its determination were presented. The operators were previously calibrated by examining ten teeth that did not qualify for the research. Kappa statistics were used to test the inter-rater reliability. The kappa for the internal consistency of the fieldwork raters was >0.87. These experts were blinded to each other’s selections. To ensure double blinding, the bleaching protocol was performed in a different room from where the evaluator examined the patients, the randomization was alphanumerically coded to ensure blinding of the research team. If all three operators chose the same shade, the selected shade was the correct shade for that tooth. If two operators had the same selection and the other one operator chose a different, the majority color was considered the precise shade. A consensus opinion of the experts was sought to agree on the correct shade if all three operators chose three different shades.

Two different concentrations of the bleaching agent were used in the research: 30% carbamide peroxide VivaStyle 30% (Ivoclar Vivadent, Schaan, Liechtenstein), and 35% hydrogen peroxide Opalescence Endo 35% (Ultradent, South Jordan, Utah, USA).

The teeth were divided into three groups, depending on the bleaching technique: walking bleach technique (10 patients), in-office technique (10 patients), and combined technique (10 patients). The teeth were selected by random sampling method.

In the walking bleach technique, ten teeth are divided into two groups. The first group consisted of five teeth treated with 30% carbamide peroxide, and the second group consisted of five teeth treated with 35% hydrogen peroxide. The teeth were isolated with dental paper rolls and a suction cup. The gingiva and the surrounding mucosa were covered with the protective agent OpalDam (Ultradent, South Jordan, Utah, USA). An access cavity was formed on the oral surface. The bleaching agent was applied in the pulp chamber. The excess fluid was removed with dry cotton balls [2,9,17]. The teeth were then sealed with glass ionomer cement Fuji IX (GC, Japan) until the next treatment.

For in-office bleaching, the external and internal tooth bleaching was performed at the same time. Before the treatment itself, the gingiva and the surrounding mucosa were covered with the protective agent OpalDam (Ultradent, South Jordan, Utah, USA). The procedure was performed on ten teeth, divided into two groups. The first group consisted of five teeth treated with 30% hydrogen peroxide, and the second group consisted of five teeth treated with 35% carbamide peroxide. An access cavity was formed on the oral surface. The bleaching agent was placed on the vestibular surface of the tooth and into the tooth cavity. The bleaching agent was left to sit on the tooth for 15 min, then removed with an aspirator and washed out with water. A cotton ball was placed into the cavity and the tooth was sealed with a temporary filling material FUJI IX, until the next session.

The combined bleaching technique included the application of in-office bleaching and the walking bleach technique. The procedure was performed on ten teeth, divided into two groups. The 30% carbamide was applied to the vestibular surface of one-half of the teeth (five teeth), while the 35% hydrogen peroxide was applied to the remaining five teeth for 15 min. The teeth were isolated with dental paper rolls and the gingiva and the surrounding mucosa were covered with the OpalDam. Upon the completed external bleaching, the same agent was applied into the cavity created on the oral surface of the tooth and left to react. The cavity was sealed with a temporary filling and a new bleaching procedure was carried out four days later.

The bleaching procedure was repeated in all the patients three times [9,18]. 

At the end of the treatment, a calcium hydroxide paste Calyx, (VOCO, Cuxhaven, Germany) was placed into each tooth and left there for 14 days to neutralize the acidic environment and prevent root resorption. The composite material Tetric EvoCeram (Ivoclar Vivadent, Schaan, Liechtenstein) was used for the definitive restoration of the cavity.

Upon the completed treatment, the new color was determined for all the teeth based on a Vita shade guide. 

For statistical data analysis, the bleached teeth were analyzed based on the modified scale from a study by Ari et al. [19].

0—the tooth was not bleached;

1—the tooth was insufficiently bleached;

2—the tooth was bleached, yet the desired color was not achieved;

3—the tooth was successfully bleached, and the desired color was achieved.

Number 0 (the tooth was not bleached) was assigned to a tooth whose color has remained unchanged after the bleaching process.

Number 1 (the tooth was sufficiently bleached) was assigned to a tooth with a slight color change, the color remained similar to the color of the discolored tooth, 1–2 shades of color on the Vita Classic guide.

Number 2 (the tooth was bleached, yet the desired color was not achieved), was assigned to the tooth where the bleaching process was successful, but the discolored tooth had not returned to its original shade, 3–4 shades of color on the Vita Classic guide.

Number 3 (the tooth was successfully bleached, and the desired color was achieved) was assigned to a tooth that has been successfully bleached and the discolored tooth had returned to its original shade, the same shade of color on the Vita Classic guide.

### 2.6. Statistical Analysis

The statistical analysis was performed using JASP 0.15.0.0. 

Using the Shapiro–Wilk test, we tested the normality of the distributions on both measuring scales of tooth color. On the measuring scale going from 0 to 3, the test result was W = 0.67 (*p* < 0.001), and on the measuring scale going from 1 to 16 for the distribution of the scores before teeth bleaching, the test result was W = 0.834 (*p* < 0.001), while for the distribution of scores after the teeth bleaching, the test result was W = 0.86 (*p* = 0.001). The test of normality of the distribution in all three cases showed that the score distributions deviate significantly from normality. Due to the statistically significant deviation of the distributions from the normal, but also due to the nature of the variables, i.e., the measurement scales and the sample size, we had to statistically process the bleaching effects for all three techniques using non-parametric tests. Χ^2^, Kruskal–Wallis, and Dunn’s post hoc comparisons tests were used. Using the Χ^2^ test, we tested the statistical significance of differences in the frequencies of certain data (the frequency of certain subgroups of patients in relation to the frequency of bleaching quality ratings). Due to the insufficient discrimination of certain subgroups of patients, as well as evaluations of the quality of bleaching, we were forced to condense two or three categories of data into one, which is visible in some contingency tables. The Kruskal–Wallis test was used to test the significance of the differences between the three bleaching techniques, but in relation to another measuring scale of tooth color (1–16). Dunn’s post hoc comparisons test was used to compare pairs of bleaching techniques. 

Significance was recognized when *p* < 0.05; the significance level was 5% (alpha = 0.05). All results are presented numerically and tabularly.

## 3. Results

A total of 26 patients participated in the clinical part of the research, including 9 men and 17 women. The youngest patient was 21 years, and the oldest was 47 years. The average age was 31.3 years. All patients were Caucasian.

### 3.1. Bleaching Results Based on the Time Elapsed after Endodontic Treatment

The Χ^2^ test established a statistically significant difference between the teeth that had been endodontically treated in the previous 5 years and the teeth that had been endodontically treated more than 5 years before (Χ^2^ = 13.659; df = 1; *p*< 0.001) (Table 1).

### 3.2. Bleaching Results Depending on Etiological Factors

Based on the Χ^2^ test results (Χ^2^ = 3.240; df = 3; *p* = 0.356), no statistically significant difference (*p* > 0.05) was established in the success of bleaching between the etiological groups (Table 2).

### 3.3. Bleaching Results Depending on the Change in Color

Kruskal–Wallis test (KWT = 2.667; df = 2; *p* = 0.236) showed no statistically significant difference in relation to the walking bleach, in-office, and combined techniques (Table 3). 

Dunn’s post hoc comparisons of pairs of techniques additionally proved that this difference was not statistically significant in all three comparisons (In-office—Combined, *p* = 0.343; In-office—Walking Bleach, *p* = 0.121; Combined—Walking Bleach, *p* = 0.058) (Table 4).

### 3.4. Bleaching Results Depending on the Bleaching Agent and Technique

The Χ^2^ test confirmed no statistically significant difference between the bleaching techniques (Χ^2^ = 1.148; df = 2; *p* = 0.563) (Table 5).

The Χ^2^ test showed no statistically significant difference in relation to the bleaching techniques (walking bleach, in-office, combined), nor in relation to the bleaching agents tested (30% carbamide peroxide and 35% hydrogen peroxide) (*p* > 0.05) (Table 6).

## 4. Discussion

The first null hypothesis was not confirmed because a significant difference was found between the effectiveness of bleaching and the time elapsed after endodontic treatment. Tooth bleaching was the most efficient in the patients who had an endodontic treatment in the past 5 years.

An important factor in the success of the bleaching treatment is the time the discoloration occurred. A shorter time of tooth discoloration has a better prognosis of bleaching, whereas the outcome of bleaching older discolorations is uncertain and oftentimes unsuccessful due to a more complex integration of color in the dentin [2]. The results of our research are in line with these facts. In previously endodontically treated teeth, the bleaching success depended on the time since endodontic treatment. In this randomized clinical study tooth bleaching was the most efficient in the teeth who had an endodontic treatment in the past 5 years (15 teeth), whereas the teeth treated more than 5 years before (4 teeth) were bleached less successfully.

Our study confirms that there was no statistical dependence on the success of tooth bleaching in relation to the etiological factor. Nevertheless, observing the individual results, the lowest success of bleaching was in the teeth whose discoloration was caused by endo sealer, which can be explained by the fact that it was a discoloration caused by an artificial source less capable of the chemical reaction of oxidization which leads to molecular conversion [2,10,20]. It has been proven that the discoloration caused by metal ions (silver ions in endo sealer, metal oxidation, metal stains) is less likely to be successfully bleached [2,10,20,21]. Nevertheless, Feiz et al. have demonstrated that teeth discolored by the AH 26 root canal sealer may successfully be bleached with carbamide peroxide, independently or combined with sodium perborate. They ascribe the success of bleaching to high concentrations of carbamide peroxide, as well as the relatively new tooth discolorations, which are easier to bleach compared to older discolorations [22].

The second, third, and fourth hypotheses were confirmed, as no significant difference was established between the bleaching efficiency and the discoloration cause, and neither was it established between the bleaching efficiency and the technique, nor the bleaching agent selected.

In our research, the best results were for teeth with an unknown cause of discoloration. In the unknown cause of discoloration group, we included the teeth in which we were unable to define the cause of the discoloration clearly, as, apart from the history of trauma, we observed traces of endo sealer and gutta percha in the cavity. Endodontic filling materials frequently left inside the chamber or pulp horns lead to tooth discoloration. Removing the excess material can change the tooth color. Removing two millimeters of the material below the cementoenamel junction and placing a glass ionomer cement barrier allows for adequate intracoronal bleaching [23]. This is how we removed the possible cause of discoloration and ensured a better penetration of the bleaching agent through dentinal tubules. The cause of tooth discoloration after trauma are pigments of natural origin. Research has shown that pigments originating from biological compounds enter a chemical reaction of dissolution more easily than artificial pigments, so their conversion takes less time than that of the pigments of artificial origin [2,10,19,21].

Research by Savić Stanković has shown independence between the etiological factor (endo sealer, trauma, necrosis, unknown cause) and the final treatment outcome [2]. The greatest success was achieved by bleaching the teeth discolored as a consequence of necrosis, which may be explained by the fact that the products of decaying dental pulp, as well as bacterial pigments, have a greater molecular conversion in relation to the pigments generated by degeneration of blood components. There was no statistically significant difference between the teeth with unknown etiological factors and the other groups of teeth, so the author supposed it was the group with several causes of discoloration [2].

Looking at the success of bleaching depending on the average bleaching score, there was no statistically significant difference between the walking bleach, in-office, and combined bleaching technique. However, looking at individual scores of bleaching success by techniques, it can be observed that the best efficiency was achieved using the in-office technique (eight of the teeth successfully bleached and the desired color was achieved), unlike the walking bleach technique (seven of the teeth were successfully bleached and the desired color was achieved) and the combined technique (five of the teeth were successfully bleached and the desired color was achieved).

A study by Santos et al. has shown equal efficiency of bleaching non-vital teeth using the in-office and walking technique of bleaching discolored bovine teeth, as well as the efficiency of all of the three agents used [24]. However, the authors give precedence to the in-office technique of bleaching using hydrogen peroxide due to the shorter time required to achieve the desired color, unlike with carbamide peroxide and sodium perborate which required more sessions to remove the existing tooth discoloration [24].

Bizghan et al. conducted a clinical study of bleaching non-vital teeth using an at-home technique and 10% carbamide peroxide, intracoronal bleaching technique with a mixture of sodium perborate and 3% hydrogen peroxide, and the so-called modified walking bleach technique, including an open access cavity and application of a splint for at-home bleaching with 10% carbamide peroxide [25]. The modified technique showed the best bleaching results immediately after the completion of bleaching, whereas the conventional technique came in second. The authors found that six months after the treatment both these techniques were equally efficient. However, they gave precedence to the modified bleaching technique due to less time necessary to achieve the desired results, the comfort of treatment at home, reduced appointment time, and the possibility of the patient’s independent control of color change [25].

In our study, the teeth were bleached using two agents, 30% carbamide peroxide and 35% hydrogen peroxide. The 30% carbamide peroxide is equivalent to 10.5% hydrogen peroxide. Even though we expected the 35% hydrogen peroxide, the strongest concentration of the agent in the research, to be more successful, the results show equal efficiency of carbamide peroxide and hydrogen peroxide. This situation can be explained by the fact that hydrogen peroxide has an excess active substance that is not reactive in the process of bleaching; carbamide peroxide has a high pH, and better penetrates into dentinal tubules [2,26]. Carbamide peroxide in situ disintegrates into urea, ammonia, carbon dioxide, water, and hydrogen peroxide, which is an active substance. Urea remarkably penetrates the enamel, affects the interprismatic region, and denaturalizes the proteins, producing a structural change in enamel and dentin. The action of urea on hard dental tissues enables better permeability for hydrogen peroxide and free radicals [1,26].

Lim et al. obtained the same results [25]. These researchers believe that carbamide peroxide and hydrogen peroxide were equally efficient because of the excess hydrogen peroxide spreading through the tissue, better penetration of carbamide peroxide through the dentin, and the relationship between the pH and the speed of the bleaching reaction. The higher the pH of carbamide peroxide, the more available free radicals for bleaching. Given that the 35% hydrogen peroxide has a pH of 3.7, and that the 35% carbamide peroxide has a pH of 6.5, carbamide peroxide may have an equivalent quantity of free radicals available for the bleaching process as the 35% hydrogen peroxide [26].

Research by Behl et al. has also confirmed equal efficiency of carbamide peroxide and hydrogen peroxide, as well as of sodium perborate in bleaching discolored teeth [27]. The good efficiency of carbamide peroxide may be ascribed to its ability to improve permeability and penetrate deeper through the tissue, allowing a reversal in the chromatic discoloration of dental tissue by oxidation reactions [28]. Similar results were obtained in the research by Bersezio et al., where the 35% hydrogen peroxide and 37% carbamide peroxide showed a high level of success in bleaching discolored teeth using the walking bleach technique [28].

The strength of our research lies in the fact that this in vivo study tests the efficiency of tooth bleaching using three different techniques and two different agents, whereas the majority of studies are based on in vitro research [24,25,29]. In addition, the literature is scarce regarding the success of tooth bleaching depending on oral hygiene and dietary habits of patients, and the effect of the period of the occurrence of tooth discoloration.

Color assessment in most studies is performed using shade guide units (DSGU) [2,16,30]. The degree of color change in this study was determined based on a modified scale scored from 0 to 3. A small number of studies [19] used this scale for color assessment. This kind of color evaluation was our choice because in this way we could also assess the subjective satisfaction of the patient. For example, a score of 2 refers to a tooth that has been successfully bleached, but the original color has not been achieved. For the patient themself, this is a great success because the treated tooth with the new color is approximately the same as the original tooth color. It is a satisfactory result that positively affects the psycho-social state and quality of life of the patient, which can also be the subject of future research.

A limitation of this study is the potential subjectivity of assessing tooth color using a Vita shade guide. The shade guide is a visual comparison of the natural tooth color with shade samples from a certain manufacturer. It is a subjective method of color measurement. This type of color determination was most frequently applied in numerous studies that monitored longitudinal changes during the bleaching treatment [2,25,26,29]. Several factors can influence the validity of the results of color determination using this method, such as the lighting in the room, experience, age, eye fatigue, and individual perception of the color [2,16,30,31,32]. Meireles et al. have found that the percentage of sensitivity and specificity of the visual assessment of color using a Vita shade guide was 86.9% and 81.9%, respectively, against the gold standard of a spectrophotometer. The authors believe that, although subjective, the visual assessment using the Vitapan Classical guide is a valid and reliable method for differentiating between dark and light shades [30]. With respect to these facts, colors were analyzed in this research by three dentists at the same time, and the determined color was agreed upon by at least two dentists.

## 5. Conclusions

The effectiveness of non-vital tooth whitening is affected by the time elapsed after endodontic treatment.

However, there was no statistically significant difference in bleaching efficiency concerning the cause of discoloration or bleaching techniques tested (walking bleach, in-office, combined), nor with respect to the bleaching agents tested (30% carbamide peroxide and 35% hydrogen peroxide).

## Figures and Tables

**Figure 1 medicina-59-00018-f001:**
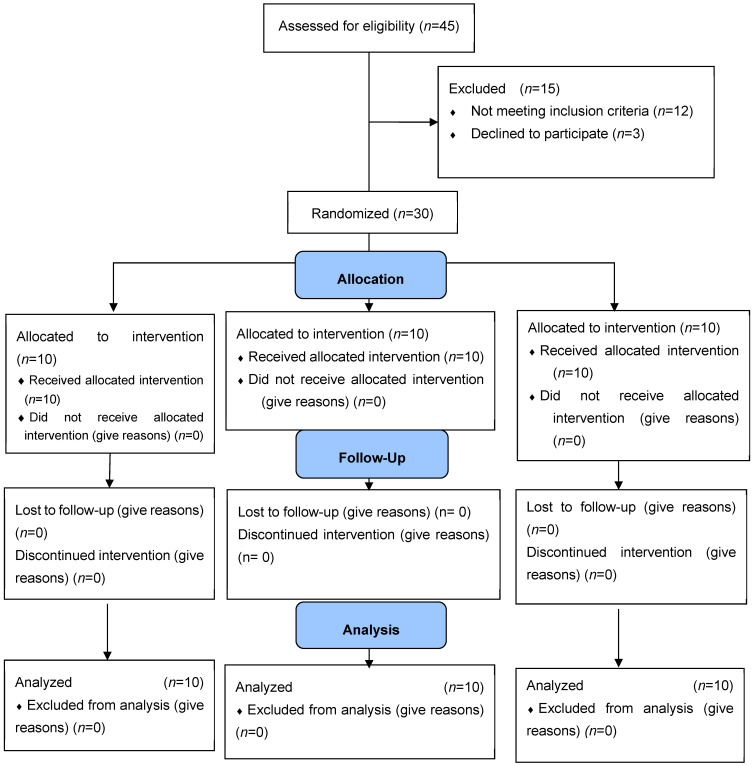
CONSORT flow diagram.

**Table 1 medicina-59-00018-t001:** Bleaching success depending on the time elapsed after endodontic treatment.

The Time Elapsed after Endodontic Treatment		Score				Total	Statistical Significance
		0	1	2	3		
0–5 years	*n*	**1**			15	16	*p* < 0.05
>5 years	*n*	10			4	14	
Total	*n*	11			19	30	

*n*—number of teeth, *p*—statistical significance, 0—the tooth was not bleached, 1—the tooth was insufficiently bleached, 2—the tooth was bleached, yet the desired color was not achieved, 3—the tooth was successfully bleached, and the desired color was achieved.

**Table 2 medicina-59-00018-t002:** Bleaching success depending on etiological factors.

		Score				Total	Statistical Significance
Etiological Factor		0	1	2	3		
Endo sealer	*n*	3			4	7	0.356
Necrosis/infection	*n*	6			6	12	
Unknown	*n*	0			3	3	
Trauma	*n*	2			6	8	
Total	*N*	11			19	30	

*n*—number of teeth, *p*—statistical significance, 0—the tooth was not bleached, 1—the tooth was insufficiently bleached, 2—the tooth was bleached, yet the desired color not was achieved, 3—the tooth was successfully bleached, and the desired color was achieved.

**Table 3 medicina-59-00018-t003:** The success of bleaching depending on the change in color.

	Technique		
	In-Office	Combined	Walking Bleach
Valid	10	10	10
Missing	0	0	0
Median	5.000	6.000	4.000
Minimum	5.000	5.000	2.000
Maximum	13.000	12.000	15.000

**Table 4 medicina-59-00018-t004:** Comparison of pair of bleaching techniques.

Comparison	z	W_i_	W_j_	*p*	*p* _bonf_	*p* _holm_
In-office—Combined	−0.403	16.450	17.950	0.343	1.000	0.343
In-office—Walking bleach	1.169	16.450	12.100	0.121	0.364	0.242
Combined—Walking bleach	1.572	17.950	12.100	0.058	0.174	0.174

z—Test-statistic for each pairwise comparison, *p*
_bonf_—significance of the difference according to Bonferroni, *p*
_holm_—significance of the difference according to Holm, W_i_—the sum of the ranks of the first technique in the comparison, W_j_—the sum of the ranks of the second technique in the comparison, *p*—statistical significance.

**Table 5 medicina-59-00018-t005:** Bleaching results depending on the bleaching technique.

Technique		Score				Total	Statistical Significance
		0	1	2	3		
In-office	* **n** *	3			7	10	*p* = 0.563
Combined	* **n** *	5			5	10	
Walking bleach	* **n** *	3			7	10	
Total	* **n** *	11			19	30	

*n*—number of teeth, *p*—statistical significance, 0—the tooth was not bleached, 1—the tooth was insufficiently bleached, 2 –the tooth was bleached, yet the desired color not was achieved, 3—the tooth was successfully bleached, and the desired color was achieved.

**Table 6 medicina-59-00018-t006:** Bleaching success depending on the bleaching agent and technique.

Concentration	Technique	Score			Total		Statistical Significance
		0	1	2	3		
30% carbamide peroxide	In-office	2			3	5	*p* = 0.563
	Combined	2			3	5	
	Walking bleach	2			3	5	
	Total	7			8	15	
35% hydrogen peroxide	In-office	0			5	5	
	Combined	3			2	5	
	Walking bleach	1			4	5	
	Total	4			11	15	

*p*—statistical significance, 0—the tooth was not bleached, 1—the tooth was insufficiently bleached, 2 –the tooth was bleached, yet the desired color not was achieved, 3—the tooth was successfully bleached, and the desired color was achieved.

## Data Availability

The data that support the findings of this study are available upon reasonable request from the corresponding author.

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
