# Peer review of "Clinical Testing of Walking Bleach, In-Office, and Combined Bleaching of Endodontically Treated Teeth"

_medicina, 2022, doi:10.3390/medicina59010018_

Round 1

Reviewer 1 Report

I praise the work of the authors, however, there are many flaws in the methodology. Moreover, nowadays is rare for bleaching studies not to provide any objective measurements. Even if visual analysis is the only tooth colour assessment method applied, there are some methods to objectevly measure the data recovered - not doing this is not a study limitation, is a authors' choice. 

Nevertheless, the theme of the paper is interesting and some interesting ideas were written in the discussion section. Therefore, I would recommend making a major revision to the methodology/results of this study in order to provide better value to it. As of now, is far from being suitable for publication.

- There is no null hypothesis established

- It's usual for clinical trials to be submitted to a clinical studies database (e.g. ClinicalTrials). This should be stated in text. If, for some particulary reason this was not made, it should be also mentioned.

- 30 patients were selected based on what? Authors should always provide a sample size paragraph explaining how they came to the final sample number (and the data they based on: pilot study? literature data? which SD? which alpha/beta considered) etc.)

- A study design should be provided

- All inclusion criteria are exclusion criteria. Inclusion criteria refers to patient's characteristics that needs to be included (e.g. having more than 18 years old, accepting to sign informed consent, accepting to interrupt soming habits etc.). Exclusion criteria refers to patient's confounding variables that should be removed from equation.

- "...according to the American Dental Association guideline" - this should have a reference.

- It should be mentioned how many operators the experience/training of each operator

- Bleaching protocols were based on what? Manufacturer's instruction, literature data (application of Calyx would probabbly fall into this category and references shoud be provided?) This should be mentioned. Additionally for the walking bleach technique how was isolation performed? Rubber dam?

- The scale provided to evaluate bleaching efficacy does not provide objective thresholds. What is a "slight color change"? What is a "successfully bleaching"? It is two shades difference on VITA Classical, three, four etc. 
If this is not establish all analysis is based on highly subjective evaluations. In 2022 is already rare to see bleaching studies without objective measurements (spectrophotometer, photography, providing CIELAB values etc.), if visual analysis interpretation is also subjective the value of the paper goes down a lot. The authors mentioned this as a limitation in the discussion, but is not actually a limitation since they could simply use another method for visual analysis interpretation: e.g. converting VITA shades into CIELAB values and calculating respective colour differences (some studies already provide this methodology), use DSU values and establish which indicates bleaching efficacy

- There is no reference on how the results will be presented (is it media, median? SD, confidence interval?). Additionally there is no analysis of normality provided, no reason why non-parametric tests were chosen etc... Statistical analysis needs a major revision. Moreover, if significance was consideraded at p<0.05 it should be mentioned that significance level was 5% or alpha = 0.05. 

- Instead of providing percentages for how many man/woman it should be exact numbers. Additionally, demographic data is poorly presented.

- The use of statistically tests is completely wrong. Mann-Whitney U is used for two variables and is being used for three. Tests like Wilcoxon should be considered.

- English needs to be revised, some sentences are hard to understand, have mistakes or both (e.g.; "...in healthy patientes on the of their dental status...". Additionally there some words written in UK English and other in US English, one of them should be decided for the entire paper.

Author Response

Review report 1

I praise the work of the authors, however, there are many flaws in the methodology. Moreover, nowadays is rare for bleaching studies not to provide any objective measurements. Even if visual analysis is the only tooth colour assessment method applied, there are some methods to objectevly measure the data recovered - not doing this is not a study limitation, is a authors' choice. 

Nevertheless, the theme of the paper is interesting and some interesting ideas were written in the discussion section. Therefore, I would recommend making a major revision to the methodology/results of this study in order to provide better value to it. As of now, is far from being suitable for publication.

Point 1: There is no null hypothesis established

Response 1:

 I have added the null hypotheses to the paper.

The null hypotheses were:

  1. There is no difference in clinical efficiency of intracoronal bleaching between the time elapsed after endodontic treatment.
  2. There is no difference in clinical efficiency of intracoronal bleaching between different discoloration causes.
  3. There is no difference in clinical efficiency of intracoronal bleaching between different bleaching techniques.
  4. There is no difference in clinical efficiency of intracoronal bleaching between different bleaching agents.

Point 2:  It's usual for clinical trials to be submitted to a clinical studies database (e.g. ClinicalTrials). This should be stated in text. If, for some particulary reason this was not made, it should be also mentioned.

Response 2:

At the time of the finalization of this research, it was not necessary to register the study in the database of clinical investigations. Subsequent registration of the study was not possible.

Point 3: 30 patients were selected based on what? Authors should always provide a sample size paragraph explaining how they came to the final sample number (and the data they based on: pilot study? literature data? which SD? which alpha/beta considered) etc.)

Response 3: I have added the sample size calculation:

A power analysis was completed for a repeated measures analysis of variance. The computed effect size for the change in color (ΔE) was found to be 0.7, using an alpha (α) level of 5% and a beta (β) level of 10%, i.e., power = 90%. This study included eight cases per group for a total of 24 cases. To compensate for a drop-out rate of approximately 15%-20%, the required total sample size in this study will be 30 cases. A sample size calculation was performed using G*Power® Version 3.1.9.7.

Point 4: A study design should be provided

Response 4: I have corrected it and provided the study design:

This blind randomized clinical study compared three methods of intracoronal bleaching. Patients were invited to participate in through examinations at the exercises of dental students.

The research involved 30 endodontically treated teeth in 26 healthy patients. Three teeth were bleached in one patient, two endodontically treated teeth in two patients each, and one tooth bleached in 23 patients. The patients were selected based on their dental status, namely the presence of discolorations in non-vital and endodontically treated teeth.

Inclusion criteria were: male and female patients between 18 and 50 years of age who signed up to participate in this research, with good systemic and oral health and visibly discolored, non-vital or endodontically treated incisors, canines and premolars, colors darker than A2 on the Vita Classical shade guide.

The exclusion criteria were: advanced periodontal diseases, teeth with large fillings or large carious lesions that could not be reconstructed using conservative procedures, tooth pain, periapical lesions, bone resorption and systemic diseases, pregnant or lactating, moderate or severe fluorosis, tetracycline stains, orthodontic treatment, orofacial tumors, trauma, or tooth malformation.

All participants were divided into three groups, 10 teeth in each group according to the technique of bleaching system adopted: first group – walking bleach technique; second group – in-office bleaching gel; third group –combination walking bleach and in-office bleaching techniques.

Point 5: All inclusion criteria are exclusion criteria. Inclusion criteria refers to patient's characteristics that needs to be included (e.g. having more than 18 years old, accepting to sign informed consent, accepting to interrupt soming habits etc.). Exclusion criteria refers to patient's confounding variables that should be removed from equation.

Response 5: I have changed the inclusion and exclusion criteria.

 Point 6: "...according to the American Dental Association guideline" - this should have a reference.

Response 6: I have added the reference.

- Point 7: It should be mentioned how many operators the experience/training of each operator

Response 7: I have added: The operators were professors of dentistry. Before this research, they attended training on the basic concepts of color and the factors that may influence its determination were presented. The operators were previously calibrated by examining ten teeth that did not qualify for the research.

- Point 8: Bleaching protocols were based on what? Manufacturer's instruction, literature data (application of Calyx would probably fall into this category and references should be provided?) This should be mentioned. Additionally for the walking bleach technique how was isolation performed? Rubber dam?

Response 8: I have added references and how the isolation had been performed:

The teeth were isolated with dental paper rolls and the gingiva and the surrounding mucosa were covered with the OpalDam.

- Point 9: The scale provided to evaluate bleaching efficacy does not provide objective thresholds. What is a "slight color change"? What is a "successfully bleaching"? It is two shades difference on VITA Classical, three, four etc. 
If this is not established all analysis is based on highly subjective evaluations. In 2022 is already rare to see bleaching studies without objective measurements (spectrophotometer, photography, providing CIELAB values etc.), if visual analysis interpretation is also subjective the value of the paper goes down a lot. The authors mentioned this as a limitation in the discussion, but is not actually a limitation since they could simply use another method for visual analysis interpretation: e.g. converting VITA shades into CIELAB values and calculating respective colour differences (some studies already provide this methodology), use DSU values and establish which indicates bleaching efficacy

Response 9: I have changed that:

Number 1 (the tooth is sufficiently bleached) is assigned to a tooth with a slight color change, the color remained similar to the color of the discolored tooth, 1-2 shade of color on the Vita Classic guide.

Number 2 (the tooth bleached, yet the desired color was not achieved), is assigned to the tooth where the bleaching process was successful, but the discolored tooth had not returned to its original shade, 3-4 shade of color on the Vita Classic guide.

Number 3 (a tooth successfully bleached, the desired color was achieved) is assigned to a tooth that has been successfully bleached and the discolored tooth had returned to its original shade, same shade of color on the Vita Classic guide.

Point 10:  There is no reference on how the results will be presented (is it media, median? SD, confidence interval?). Additionally there is no analysis of normality provided, no reason why non-parametric tests were chosen etc... Statistical analysis needs a major revision. Moreover, if significance was consideraded at p<0.05 it should be mentioned that significance level was 5% or alpha = 0.05. 

Response 10: I have changed the statistical analysis, and I have added new non-parametric tests, alpha, SD, ΔSGU…

Point 11:  Instead of providing percentages for how many man/woman it should be exact numbers. Additionally, demographic data is poorly presented.

Response 11: I have changed it to:

A total of 26 patients participated in the clinical part of the research, including 9 men and 17 women. The youngest patient was 21, and the oldest was 47. The average age was 31.3. All patients were Caucasian.

Point  12: The use of statistically tests is completely wrong. Mann-Whitney U is used for two variables and is being used for three. Tests like Wilcoxon should be considered.

Response 12: Due to the insufficient discrimination of certain subgroups of patients, as well as evaluations of the quality of bleaching, we were forced to condense two or three categories of data into one, which is visible in some contingency tables. I have changed statistical analyses in Χ²  and Kruskal-Wallis test.

Point  13: English needs to be revised, some sentences are hard to understand, have mistakes or both (e.g.; "...in healthy patientes on the of their dental status...". Additionally there some words written in UK English and other in US English, one of them should be decided for the entire paper.

Response 13: The English has been revised.

Reviewer 2 Report

This is a well-developed article on an interesting topic for today’s dentistry. However, some issues need to be addressed before considering it for publication:

There are some typos that need to be checked. Review point "2.1. Creating the Research Sample" Paragraph must be rewritten. Also, in the discussion section, some references are cited in superscript form (not in parentheses as in the full text).

Most "inclusion criteria" are not such a thing, but exclusion criteria. In fact, most of them are also repeated as exclusion criteria. It seems that the inclusion criterion was a patient with at least one non-vital disolored tooth. Also, why were patients taking analgesic, anti-inflammatory, or antibiotic drugs excluded? How might these drugs affect the study outcomes?

Please review the descriptions of the treatments, since for both the walking bleach technique and the in-office technique it says that all groups were treated with carbamide peroxide in concentrations of 30 or 35% (“The first group consisted of 5 teeth treated with 30% carbamide peroxide, and the second group consisted of 5 teeth treated with 35% carbamide peroxide”)

In Table 1, order the p values. It is not understood what they represent. Also in this table, the sum of the participants for each group is 27 and should be 30. In which group were the missing participants?

Finally, results must be taken carefully considering that the only statistically significant difference was when considering the time when endodontic treatment was completed, but only 3 participants had more than 10 years. This should be clearly stated

Author Response

Review report response

This is a well-developed article on an interesting topic for today’s dentistry. However, some issues need to be addressed before considering it for publication:

Point 1: There are some typos that need to be checked. Review point "2.1. Creating the Research Sample" Paragraph must be rewritten. Also, in the discussion section, some references are cited in superscript form (not in parentheses as in the full text).

Response 1: I have changed the paragraph “Creating the Research Sample” to “Study design”.

Point 2: Most "inclusion criteria" are not such a thing, but exclusion criteria. In fact, most of them are also repeated as exclusion criteria. It seems that the inclusion criterion was a patient with at least one non-vital discolored tooth. Also, why were patients taking analgesic, anti-inflammatory, or antibiotic drugs excluded? How might these drugs affect the study outcomes?

Response 2: I have since changed the inclusion and exclusion criteria.

Point 3: Please review the descriptions of the treatments, since for both the walking bleach technique and the in-office technique it says that all groups were treated with carbamide peroxide in concentrations of 30 or 35% (“The first group consisted of 5 teeth treated with 30% carbamide peroxide, and the second group consisted of 5 teeth treated with 35% carbamide peroxide”)

Response 3: I have changed 35% carbamide peroxide to 35% hydrogen peroxide.

Point 4: In Table 1, order the p values. It is not understood what they represent. Also in this table, the sum of the participants for each group is 27 and should be 30. In which group were the missing participants?

Response 4:  I have changed Table 1.

Point 5: Finally, results must be taken carefully considering that the only statistically significant difference was when considering the time when endodontic treatment was completed, but only 3 participants had more than 10 years. This should be clearly stated

Response 5: I have changed Table 1, and the periods of endodontic treatment in two groups for more accurate statistics.

Round 2

Reviewer 1 Report

I praise authors revision since methodology and results for the study are much better presented. Nevertheless, are still some issues they need to be revised:

- If you justify the absence of trial registration with the time the research was conducted, then it should be mentioned the time of research and not only the place

- Study design should be provided also in a flowchart. I advise following CONSORT guidelines. Additionally should be provided information regarding randomization and blinding

- Sample size determination has some concerning problems. For once, using repeated measures ANOVA and ΔE for sample size does not make sense when you are using kruskall-wallis and visual shades in the research. Secondly, even if you are using ΔE, having an effect of 0.7 is even lower than the perceptibility threshold, thus any change wouldn't be clinically significant.

- Statistical testes are more appropriate than before, nonetheless, there is no evaluation of normality and no reason why non-parametric tests were chosen over parametric. 

- "...was rejected because a significant correlation was established". Rephrase the sentence since you didn't perform any correlation tests.

- English much better, just need minor spellcheck corrections.

Author Response

Point 1: I praise authors revision since methodology and results for the study are much better presented. Nevertheless, are still some issues they need to be revised:

- If you justify the absence of trial registration with the time the research was conducted, then it should be mentioned the time of research and not only the place

Response 1: I have added the time of research: The research was carried out between 2013 and 2014.

Point 2: - Study design should be provided also in a flowchart. I advise following CONSORT guidelines. Additionally should be provided information regarding randomization and blinding

Response 2: I have added CONSORT flow diagram, and information regarding randomization and blinding:

This blind randomized clinical study compared three different methods of intracoronal bleaching. Patients were invited to participate in thorough examinations at the exercises of dental students. The treated teeth were randomly selected. The distribution of bleaching techniques by groups of teeth was carried out by random drawing using Microsoft Excel 2010 (Microsoft, Redmond, Washington, USA) from the coding assigned to each participant. There were three groups: the first group was bleached with the walking bleach technique (10 teeth), the second group was bleached with the in-office technique (10 teeth),  and the third group was bleached with the combined technique (10 teeth). Two trained operators performed the bleaching treatments. The third operator, who had no contact with the patients, was responsible for conducting the randomization. To ensure double blinding, marks and packaging on bleaching agents were removed, the procedures and the instruments were standardized and randomization was alphanumericlly coded. The statistician received data tabulated in code that did not allow for the identification of the treatment applied to each group.

Point 3: - Sample size determination has some concerning problems. For once, using repeated measures ANOVA and ΔE for sample size does not make sense when you are using kruskall-wallis and visual shades in the research. Secondly, even if you are using ΔE, having an effect of 0.7 is even lower than the perceptibility threshold, thus any change wouldn't be clinically significant. Response 3: I have changed Sample size calculation:  

A power analysis was completed for repeated measures of the Wilcoxon test. The output parameters were: Noncentrality parameter δ=2.57; Critical t=1.71; Df=25.32; Total sample size n=24; and Actual power was Ap=0.8. This study included eight cases per group for a total of 24 cases. To compensate for a drop-out rate of approximately 15%-20%, the required total sample size in this study will be 30 cases. A sample size calculation was performed using G*Power® Version 3.1.9.7.

Point 4: Statistical testes are more appropriate than before, nonetheless, there is no evaluation of normality and no reason why non-parametric tests were chosen over parametric. 

Response 4: I have changed in Statistical analysis:

Using the Shapiro-Wilk test, we tested the normality of the distributions on both measuring scales of tooth color. On the measuring scale going from 0 to 3, the test result was W=0.67 (p<.001), and on the measuring scale going from 1 to 16 for the distribution of the scores before teeth bleaching, the test result was W=0.834 (p<.001), while for the distribution of scores after the teeth bleahing, the test result was W=0.86 (p=.001). The test of normality of the distribution in all three cases showed that the score distributions deviate significantly from normality. Due to the statistically significant deviation of the distributions from the normal, but also due to the nature of the variables, i.e. the measurement scales and the sample size, we had to statistically process the bleaching effects for all three techniques using non-parametric tests. Χ², Kruskal Wallis and Dunn's Post Hoc Comparisons tests were used. Using the Χ² test, we tested the statistical significance of differences in the frequencies of certain data (the frequency of certain subgroups of patients in relation to the frequency of bleaching quality ratings). Due to the insufficient discrimination of certain subgroups of patients, as well as evaluations of the quality of bleaching, we were forced to condense two or three categories of data into one, which is visible in some contingency tables. The Kruskal-Wallis test was used to test the significance of the differences between the three bleaching techniques, but in relation to another measuring scale of tooth color (1-16). Dunn's Post Hoc Comparisons test was used to compare pairs of bleaching techniques.

Point 5: - "...was rejected because a significant correlation was established". Rephrase the sentence since you didn't perform any correlation tests.

Response 5: I have rephrased the sentence: The first null hypothesis was not confirmed because a significant difference was found between the effectiveness of bleaching and the time elapsed after endodontic treatment.

Point 6: - English much better, just need minor spellcheck corrections.

Response 6: The English has been revised.